# HOSC: A Periodic Activation Function for Preserving Sharp Features in Implicit Neural Representations

## Abstract

Recently proposed methods for implicitly representing signals such as images, scenes, or geometries using coordinate-based neural network architectures often do not leverage the choice of activation functions, or do so only to a limited extent. In this paper, we introduce the Hyperbolic Oscillation function (HOSC), a novel activation function with a controllable sharpness parameter. Unlike any previous activations, HOSC has been specifically designed to better capture sudden changes in the input signal, and hence sharp or acute features of the underlying data, as well as smooth low-frequency transitions. Due to its simplicity and modularity, HOSC offers a plug-and-play functionality that can be easily incorporated into any existing method employing a neural network as a way of implicitly representing a signal. We benchmark HOSC against other popular activations in an array of general tasks, empirically showing an improvement in the quality of obtained representations, provide the mathematical motivation behind the efficacy of HOSC, and discuss its limitations.

## 1 Introduction

An increasingly common scenario in learning visual data representations is approximating a structured signal $\mathbf{s}\colon \mathbf{R}^k \to \mathbf{R}^m$ via a coordinate-based neural network $\mathbf{f}_{\boldsymbol{\theta}}$ parametrized by a set of parameters $\boldsymbol{\theta} \in \mathbf{R}^p$. These representations, known as implicit neural representations (INRs), are fully differentiable and offer numerous advantages over traditional counterparts such as meshes or pixel grids in optimization tasks, often requiring significantly less memory.

INRs are versatile in their application, capable of representing a variety of types of objects, including audio signals ($k, m = 1$), images ($k = 2$, $m = 1$ or $m = 3$), radiance fields ($k = 5$, $m = 4$), geometries ($k = 2$ or $k = 3$, $m = 1$), and parametrzied curves ($k = 1$, $m > 1$). For instance, to represent the geometry of a 3D object, one would obtain a dataset of evaluations $\mathbf{X} = \{(\mathbf{x}, \mathbf{s}(\mathbf{x}))\}$ of the signed distance function $\mathbf{s}(\mathbf{x}) = \mathrm{sdf}(\mathbf{x})$ with respect to the surface of that object, and find the values of parameters $\boldsymbol{\theta}$ that minimize the reconstruction loss:

$$\boldsymbol{\theta} = \operatorname*{argmin}_{\boldsymbol{\theta}} \ \mathbf{E}_{(\mathbf{x}, \mathbf{s}(\mathbf{x})) \sim \mathbf{X}}[\|\mathbf{f}_{\boldsymbol{\theta}}(\mathbf{x}) - \mathbf{s}(\mathbf{x})\|_2 + \Psi(\boldsymbol{\theta})] \,,$$

where $\Psi(\boldsymbol{\theta})$ denotes a regularizer. Instead of a regression, this task could also be posed as a classification problem, where the signal $\mathbf{s}(\mathbf{x})$ takes values in the discrete set $\{0, 1\}$, representing an occupancy field. In general, defining an appropriate domain, codomain, loss function, and regularization is a problem-specific research challenge.

Importantly, INRs introduce a new paradigm in training neural networks. In classical applications of neural networks, such as prediction, the goal is to approximate a function $\mathbf{f}$ given its noisy evaluations $\mathbf{f}(\mathbf{x})$ at sparsely sampled datapoints $\mathbf{x}$. One of the challenges is thus not to overfit the approximation to the noise present in the training data. On the contrary, for INRs, we assume the data is noise-free and more regularly sampled, and aim to encode this into the network's parameters, implying that in this context overfitting is actually desirable for capturing high-frequency details of the signal.

However, popular activation functions such as ReLU are biased towards capturing lower frequencies, which is beneficial in prediction tasks, but hinders their capability to accurately represent sharp

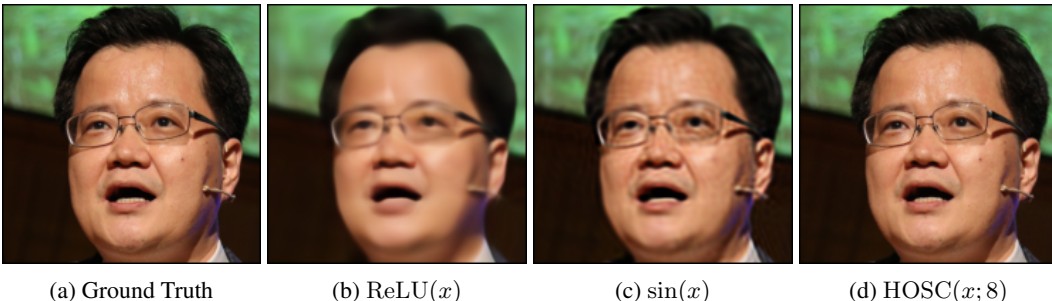

(a) Ground Truth        (b) $\text{ReLU}(x)$        (c) $\sin(x)$        (d) $\text{HOSC}(x; 8)$

Figure 1: Reconstruction of an image using an MP running different activation functions. The process involved training a five-layer coordinate-based MLP with a width of 256 for 100 iterations for each of the activations. No positional encoding and no frequency initialization has been used.

features of signals when applied as INRs. Three primary strategies to approach this problem while remaining in the INRs framework have been developed:

- **Hybrid representations.** Methods like ACORN (Martel et al., 2021), (Müller et al., 2022) and TensoRF (Chen et al., 2022) use neural networks to achieve highly detailed representations of complex signals, such as gigapixel images and radiance fields. However, they also rely on traditional data structures, and hence require storing some sort of raw data. This notably enlarges their memory footprint compared to just storing the parameters of an MLP, and results in them not being fully differentiable.

- **Positional encoding.** Fourier Feature Networks (FFNs) (Tancik et al., 2020) employ positional encoding, which has been shown to accelerate the learning of higher-frequency features. Such encodings, if sampled densely, become extremely memory inefficient, and therefore require sampling a predefined distribution. This introduces more stochasticity to the model, as well as the need to tune the distribution's parameters manually.

- **Periodic activations.** Sinusoidal Representation Networks (SIRENs) proposed by Sitzmann et al. (2020) are multi-layer perceptrons (MLPs) that utilize $\sin(x)$ instead of ReLU as their activation function. Consequently, they remain fully differentiable and offer a compact representation of the signal. While SIRENs demonstrated a significant improvement over ReLU, they struggle to capture high-frequency details in problems like shape representations, and are not-well suited for methods such as (Mildenhall et al., 2020).

In this paper, we introduce a new periodic parametric activation function — the **Hyperbolic Oscillation** activation function (**HOSC**), defined as $\text{HOSC}(x; \beta) = \tanh(\beta \sin x)$. Here, $\beta > 0$ is a controllable sharpness parameter, enabling HOSC to seamlessly transition between a smooth sinelike wave and a square signal. Similarly to SIREN, an MLP running the HOSC activation function is fully differentiable and inexpensive memorywise. However, the HOSC's sharpness parameter $\beta$ allows it to much more accurately capture sudden or sharp jumps, and hence preserve high-frequency details of the signal. Moreover, since HOSC is differentiable with respect to $\beta$, the sharpness can be adjusted automatically alongside the reset of the parameters, a method to which we refer as **Adaptive HOSC** or **AdaHOSC**.

Our extensive empirical studies show that HOSC consistently outperforms ReLU and SIREN across an array of benchmarking tasks. These tasks encompass fitting random signals, images of random square patches, photos, gigapixel images, and 2D & 3D SDF. In summary, HOSC provides an easy-to-implement method allowing simple MLPs to achieve high level of detail in signal encoding tasks without loosing differentiability or increasing memory footprint, and it does this without the need for positional encoding.

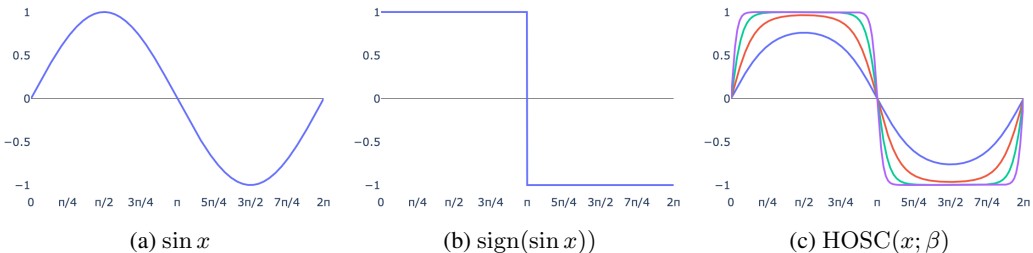

(a) $\sin x$        (b) $\text{sign}(\sin x))$        (c) $\text{HOSC}(x; \beta)$

Figure 2: Comparison of the sine, square, and HOSC waves for different values of the sharpness parameter $\beta \in \{1, 2, 4, 8\}$ and $x \in [0, 2\pi]$. As $\beta$ increases, HOSC starts to resemble a square wave.

## 2 RELATED WORK

### 2.1 IMPLICIT NEURAL REPRESENTATIONS

Currently, INRs are gaining a lot of attention in visual computing research (Xie et al., 2021). Their applications are widespread, and encompass image processing (Tancik et al., 2020), radiance fields (Mildenhall et al., 2020), 3D shape modeling (Park et al., 2019), audio and video compression (Lanzendörfer & Wattenhofer, 2023; Chen et al., 2021), physics-informed problems (Raissi et al., 2019), and solving PDEs (Sitzmann et al., 2020; Li et al., 2020). There are many reasons for choosing INRs over classical data structures:

- **Differentiability.** Given their differentiable nature, INRs offer an immediate advantage over classical, non-differentiable methods in optimization and deep learning tasks.
- **Compactness.** INRs often require less memory, as storing the parameters and hyperparameters of a neural network is typically less memory-intensive than storing raw data.
- **Continuous representation.** In principle, due to their generalization capability, neural networks enable the representation of data with arbitrary precision, making resolution a less significant issue (Chen et al., 2020).

For a more comprehensive review of the INR literature, we refer to the recent surveys by Tewari et al. (2020), Tewari et al. (2021), and Xie et al. (2021).

**Shape and geometry representation.** Classical methods of shape and geometry representation include voxel grids, polygonal meshes and point clouds. However, all of these methods suffer from limitations. Voxel grids are subject to the curse of dimensionality, which makes them inefficient in handling high-resolution data. Moreover, manipulating voxel grids and dense meshes can be computationally intensive (Xiao et al., 2020; Kato et al., 2017). Meshes are also prone to errors, and designing a mesh can be quite time-consuming for human creators. As for point clouds, they do not encode topological information (Kato et al., 2017). These issues have prompted the exploration of INRs in the context of shape and geometry modeling. The seminal work by Park et al. (2019) has demonstrated that INRs are capable of accurately representing surfaces as signed distance functions. Further research in this direction has been conducted by Atzmon & Lipman (2019), Michalkiewicz et al. (2019) and Gropp et al. (2020). Another option is presented by occupancy networks, which model the shape as the decision boundry of a binary classifier implemented as a neural network (Mescheder et al., 2018; Chen & Zhang, 2018).

**Encoding appearence.** In addition to encoding geometry, coordinate-based neural networks are also capable of representing the appearance aspects. For instance, Texture Fields (Oechsle et al., 2019) enable coloring any 3D shape based on an image. Methods such as LIIF (Chen et al., 2020) and ACORN (Martel et al., 2021) are effective in representing high-resolution gigapixel images. Furthermore, by addressing the inverse problem, Neural Radiance Fields (Mildenhall et al., 2020) allow for reconstruction of multidimensional scenes from a collection of 2D images. Other significant contribution in this area include (Müller et al., 2022; Chen et al., 2023a; 2022; Martel et al., 2021). A lot of these and similar methods are hybrid representations that combine neural networks with classical non-differentiable data structres. As such, they are not directly related to HOSC, which primarily focuses on fully differentiable architectures.

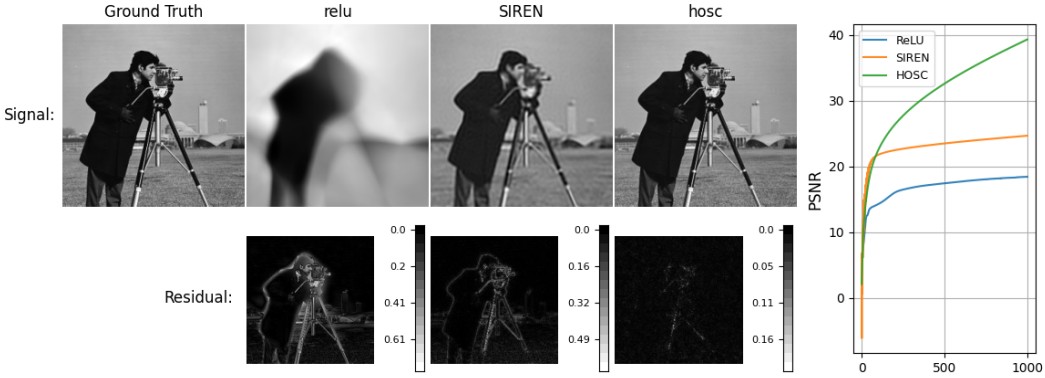

Figure 3: Comparison of an MLP running ReLU, HOSC, and a SIREN architecture fitting the cameraman image for 1000 epochs. The plot to the right shows PSNR values for each model compared to the ground truth measured at each training epoch. Below each plot the residual differences from the ground truth signal are displayed.

## 2.2 ACTIVATION FUNCTIONS AND PERIODICITY

**Activation functions.** Activations are essential for neural networks to be able to model non-linear relationships. Early activation functions include the Logistic Sigmoid, Hyperbolic Tangent, and Rectified Linear Unit (ReLU). Thanks to their low-frequency bias, they are able to deal with the noise present in the training data, possess generalization capabilities, and thus excel in applications such as prediction. In contrast to these early non-linearities, more recently proposed activation functions such as SWISH (Ramachandran et al., 2018), PReLU (He et al., 2015), SReLU (Jin et al., 2015), and MPELU (Li et al., 2016) incorporate one or more parameters, which are optimized during training along with the rest of the network's parameters. For a more in-depth survey on activation functions, refer to (Dubey et al., 2021; Apicella et al., 2020; Karlik & Olgac, 2011).

**Periodicity in neural networks.** All the non-linearities mentioned in the previous section are non-periodic. Although less common, periodic activations have been studied for many years. Early work by Sopena et al. (1999) and Wong et al. (2002) analyzed their performance in classification problems. A more recent study by Parascandolo et al. (2017) investigated which tasks are particularly well-suited to periodic activations and where they may face challenges. In (Lapedes & Farber, 1987), the authors use an MLP with sine activation for signal modeling, drawing a direct connection to the Fourier transform. A significant contribution in the field of INRs is the SIREN architecture (Sitzmann et al., 2020), which employes sine activation to solve PDEs and encode images and videos. Various aspects of periodic activations have also been studied by Ramasinghe & Lucey (2021). An alternative approach to introducing periodicity has been explored by Tancik et al. (2020), who generalize positional encoding to coordinate-based MLPs.

## 3 OUR CONTRIBUTION

### 3.1 HOSC

In this paper, we propose a novel periodic parametric activation function designed specifically for fitting INRs — the Hyperbolic Oscillation activation function, or HOSC. It is defined as

$$\mathrm{HOSC}(x; \beta) = \tanh(\beta \sin x),$$

where $\beta > 0$ is the sharpness parameter, controlling the extent to which the resulting wave resembles a square wave. This phenomenon is illustrated in Figure 2. In fact, given that $\lim_{\beta \to \infty} \tanh(\beta x) = \mathrm{sign}(x)$ for all $x \in \mathbf{R}$, we know that $\lim_{\beta \to \infty} \mathrm{HOSC}(x; \beta) = \mathrm{sign}(\sin x)$, so indeed HOSC approaches the square wave pointwise in the infinite sharpness limit. The rapid amplitude changes around $x = n\pi$ for $n \in \mathbf{N}$ at high values of $\beta$ enable HOSC to model acute features of the signal. Conversely, smooth transitions at lower $\beta$ values allow it to capture low-frequency components instead.

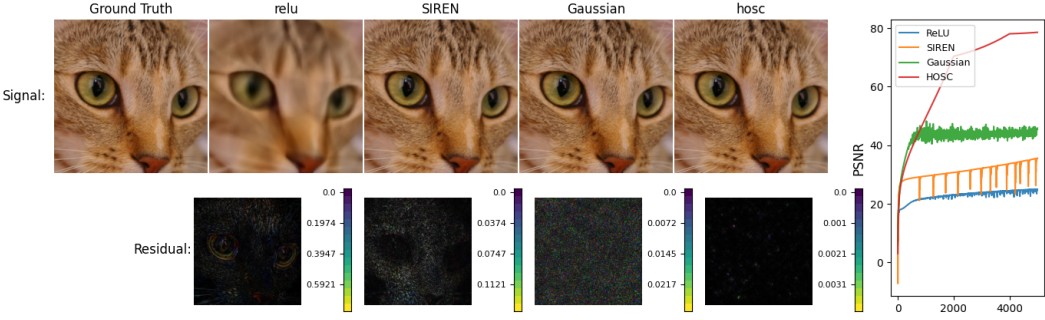

Figure 4: Comparative analysis of fitting an image of a cat for 5000 epochs. The signal reconstruction residuals are presented below the reconstruction plots, and the plot to the right shows PSNR curves for each model.

## 3.2 ADAHOSC

Importantly, HOSC is differentiable not only with respect to the input $x$, but also with respect to the sharpness parameter $\beta$:

$$\partial_\beta \text{HOSC}(x; \beta) = \sin(x)\big(1 - \text{HOSC}^2(x; \beta)\big).$$

This property allows an MLP using HOSC to optimize the sharpness parameter during training, rather than fixing it as a hyperparameter. When the sharpness factor $\beta$ is dynamically optimized, we refer to the resulting activation function as AdaHOSC, an acronym for Adaptive HOSC.

## 4 EXPERIMENTAL RESULTS

In this section, we experimentally assess the performance of HOSC in various benchmarking tests and compare it to ReLU and SIREN. More experimental results can be found in the Appendix.

### 4.1 REPRESENTING IMAGES ($\mathbf{s}\colon \mathbf{R}^2 \to \mathbf{R}$ OR $\mathbf{R}^3$)

An image can be conceptualized as a function $\mathbf{I}\colon \mathbf{R}^2 \to \mathbf{R}^n$, where $n = 1$ (for black and white images) or $n = 3$ (in case of the colored images), mapping pixel coordinates to their corresponding color intensities. To construct an INR, one commonly approximates the function $\mathbf{I}$ with an MLP, training it on all the available coordinate-color value pairs $\big((x, y), \mathbf{I}(x, y)\big)$.

In Figures 1, 3 and 4, we present the results of fitting photos with an MLP running the HOSC activation. In Figure 3, the black and white cameraman image is fitted for 1000 epochs, demonstrating that an MLP employing HOSC activation achieves a higher PSNR quicker than a ReLU-MLP or SIREN. Additionally, in Figure 4, we also fit an MLP with a Gaussian activation, previously shown to outperform SIREN (Ramasinghe & Lucey, 2021) without being sensitive to the choice of initialization scheme. We also note that we employed a linear step-wise learning rate scheduler with a rate of $\gamma = 0.1$ every 2000 epochs, as the HOSC-MLP begins to exhibit an extreme oscillatory convergence at high PSNR values without this adjustment. Although we confirm that the Gaussian activation performs better than the SIREN model in this case, our findings reveal that a HOSC-MLP achieves a significantly higher PSNR compared to both.

Figure 5 reveals more interesting results. This experiment evaluates the performance of the HOSC on images with varying frequency content. For each activation, a four-layer MLPs with a width of 256 was trained on a $256 \times 256$ black images, each containing 100 randomly placed white square patches, over 5000 epochs. Patch sizes used in the experiment are $1 \times 1$, $4 \times 4$, and $16 \times 16$. For the SIREN model, we adopted the same weight initialization and a frequency factor of 30, as detailed in (Sitzmann et al., 2020). For the HOSC-MLP, we use a sharpness factor schedule, where sharpness varies across layers with values $\beta_i = [2, 4, 8, 16]$. Additionally, a frequency factor of 30 is used in the first layer, followed by a factor of 1.0 in the subsequent layers.

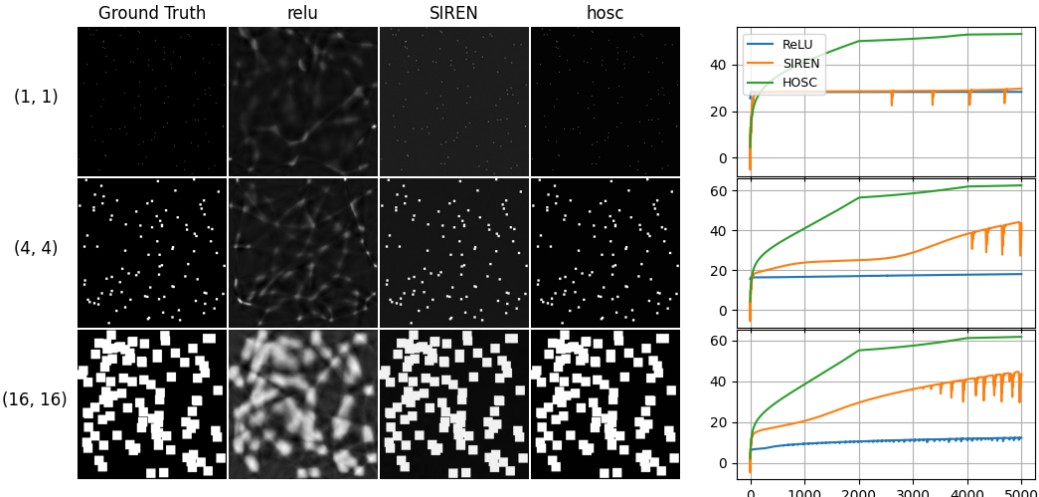

Figure 5: Comparison of a ReLU MLP, SIREN, and HOSC MLP fitting an image of random square patches of dimension 1x1, 4x4, and 16x16. The plot to the right shows PSNRs of the model to ground truth per epoch of training.

As anticipated, the ReLU network struggles to accurately capture the ground truth signal, resulting in a low converging PSNR. Interestingly, as the patch size increases, ReLU's peak PSNR decreases. In contrast, for SIREN, the peak PSNR increases as the ground truth signal losses sharp frequencies. The HOSC-MLP surpasses both of them, and is able to accurately represent the ground truth signal regardless of patch sizes. Notably, the PSNR values for the HOSC-MLP, after being trained for $5000$ epochs, significantly exceed those of both the ReLU-MLP and SIREN models.

### 4.1.1 GIGAPIXEL IMAGES

In this experiment, we compare the performances of SIREN and HOSC models on the task of Gigapixel Image Approximation, where the target signal is an RGB Image of extremely high-resolution. In this case, the SIREN model is of depth $4$, with a width of $256$, and each sinusoidal activation has a frequency of $30$. The HOSC model has the same depth and width, however, the activations have a sharpness of $\beta = 8$, while only the first activation has a frequency of $30$, and the rest has a frequency of $1$. We follow the same initialization scheme as Sitzmann et al. (2020) for both the SIREN and HOSC models.

Results of fitting both models to an image of resolution $9302 \times 8000 \times 3$ for $100$ epochs are shown in Figure 6. Although both models do not look perceptually close to the zoomed in reference photo, it is apparent that a model equipped with HOSC is able to retain sharper features for the high-resolution image whereas the SIREN essentially learns a smooth interpolation. This fact is also reinforced by analyzing the PSNR plots, where HOSC beats SIREN even at the first few epochs.

### 4.2 SDFs ($\mathbf{s} \colon \mathbf{R}^2$ OR $\mathbf{R}^3 \to \mathbf{R}$)

The signed distance $\mathrm{sdf}_\Omega(\mathbf{x})$ with respect to a shape $\Omega \subset \mathbf{R}^n$ is defined as

$$\mathrm{sdf}_\Omega(\mathbf{x}) = \begin{cases} +\rho(\mathbf{x}, \partial\Omega) & \text{if } \mathbf{x} \notin \Omega \,, \\ -\rho(\mathbf{x}, \partial\Omega) & \text{if } \mathbf{x} \in \Omega \,, \end{cases}$$

for all the coordinates $\mathbf{x} \in \mathbf{R}^n$. Consequently, by training a neural network $\mathbf{f}_\theta$ to approximate $\mathrm{sdf}_\Omega$, we can approximate the shape's boundary as the zero-level set $\{\mathbf{x} \in \mathbf{R}^n \mid \mathbf{f}_\theta(\mathbf{x}) = 0\}$.

In this section, we consider the case where $n = 2$ and $n = 3$ and train MLPs on a dataset $\{(\mathbf{x}, \mathrm{sdf}(\mathbf{x}))\}$ comprised of coordinate-SDF evaluation pairs. Our experimental exploration aims to identify any patterns that emerge when varying the depth and width of the MLPs, as well as adjusting the sharpness factor $\beta$ (2D SDF).

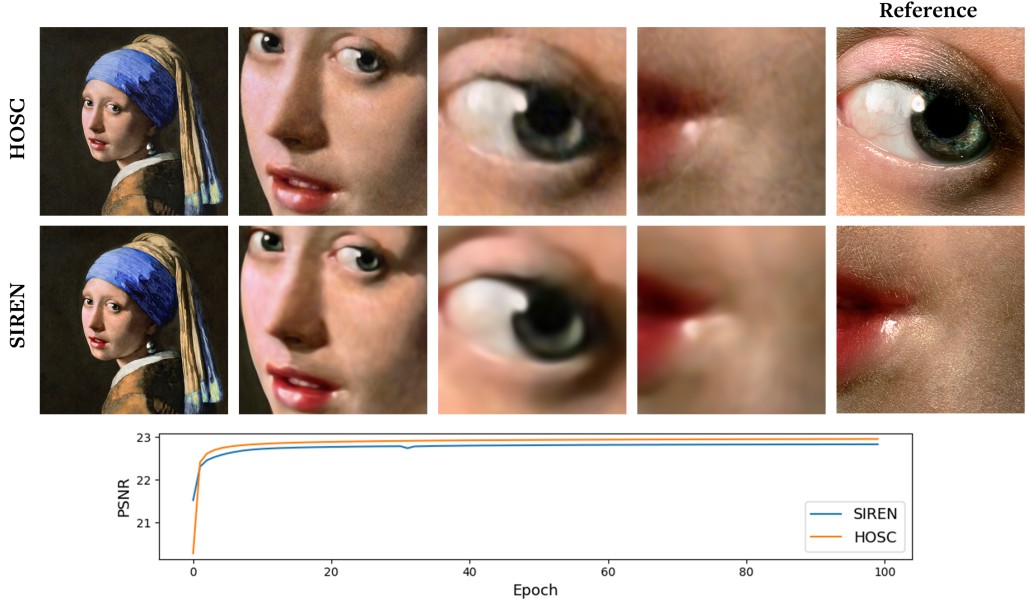

Figure 6: Top:Results of fitting a HOSC and a SIREN model to a high-resolution image. Below: a plot of PSNR per epoch for both methods.

In Figures 8 and 18 (in Appendix) we present a comparison of AdaHOSC to ReLU and SIREN. For a fair comparison, we run the same MLP architecture (5 hidden layers with 256 width) for 20 epochs only changing the activation. AdaHOSC uses the initial value of $\beta = 8$. We find that AdaHOSC provides a much higher quality of representation, as demonstrated by the IoU (intersection over union) values, suggesting faster convergence time compared to SIREN.

Moreover, the results of the 2D SDF experiment are illustrated in Figure 7. In this epxeriment, we train four-layer 512 width MLPs on 20 SDF evaluations of a regular star shape. Similar to image fitting, HOSC's performance surpasses that of ReLU and SIREN. Moreover, our findings indicate that deep HOSC-MLPs achieve higher PSNR values. Regardless of depth, it is observed that, depending on the width, greater $\beta$ values enable HOSC to more accurately represent the shape, as evidenced by the PSNR values. This observation further supports the hypothesis that HOSC can effectively represent signals with high-frequency content (including discontinuities) when the sharpness factor is large.

## 5 CONCLUSIONS AND DISCUSSION

In this paper, we have introduced the Hyperbolic Oscillation activation function $\mathrm{HOSC}(x; \beta) = \tanh(\beta \sin x)$, a new periodic parametric activation that has been designed to be particularly effective in preserving sharp features in INRs. Additionally, in Section 4, we presented experimental results that evaluate the performance of the HOSC function in comparison to existing approaches. Our findings revealed that an MLP employing the HOSC activation with a suitably chosen or automatically-optimized sharpness parameter $\beta$ consistently outperforms identical structure MLPs using ReLU and SIREN activations, and achieve the same level of accuracy in neural signal encoding problems as Fourier Feature Networks. HOSC thus offers a simple, fully differentiable and compact high-quality signals representation method with no need for hyperparameter tuning. However, we have identified scenarios where HOSC is clearly not the optimal choice, which we will explore in the following discussion.

**Spectral bias.** Different problems require a different spectral bias. While for signal encoding, where we assume that there is very little or no noise present in the data, fitting high-frequency components of the signal is advantageous. Conversely, for capturing only the general trends from a set of noisy data, a low-frequency bias can help avoid overfitting to noise. Naturally, this constraints

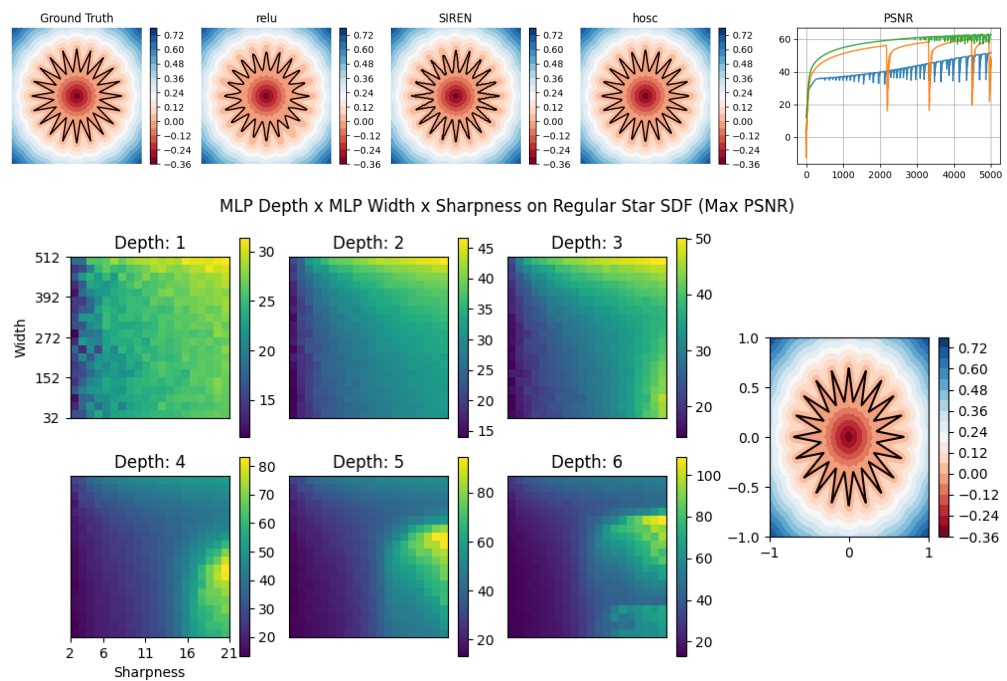

Figure 7: Comparison of coordinate-based MLPs fitting an SDF of a regular star. Top: SDFs learned by the models running different activations. Bottom: Heatmap illustrating the maximum PSNR values for HOSC-MLPs with different topologies (depth and width of layers) and sharpness factors.

the application of HOSC and other periodic activations in settings that require generalization beyond the observed datapoints. For instance, in our experiments we found that both HOSC and SIREN activations underperform when applied to Neural Radiance Fields (Mildenhall et al., 2020), which perform best with various types of positinoal encoding, either in freqnecty domain, or parametric encoding combined with spatial data strcutres (also denoted as hybrid) and ReLU activations.

**Solving PDEs.** Cooridnate-based neural networks have been applied to solving PDEs in physics (Raissi et al., 2019). However, we observe that HOSC is not particularly suited for these types of problems, compared to e.g. SIREN architecture. We attribute this limitation to the increasing complexity found in subsequent derivatives of HOSC. While the derivatives of a fully-connected SIREN layer remain SIRENs (Sitzmann et al., 2020), enabling it to accurately fit both the signal and its derivatives, the situation is more convoluted. As a result, HOSC preserves the signal's sharp features but at the expense of derivative information.

**Hybrid and parametric positional encoding.** In our experiments we apply HOSC in signal encoding scenarios, like images, giga-pixel images, and 3D SDFs, where a HOSC-MLP achieves similar reconstruction quality, however, not the timings of highly optimized methods as InstantNGP (Müller et al., 2022), ACORN (Martel et al., 2021), grid-based Dictionary Fields (Chen et al., 2023a), or TensoRF (Chen et al., 2022). There methods shorten the training and inference times at the cost of a higher memory footprint and a more sophisticated implementation. In contrast, a simple MLP is much easier to implement and storing its parameters is far less demanding in terms of memory. Finally, it is important to note that, in principle HOSC can be utilized in hybrid representations as well, whenever a coordinate-based MLP is used to overfit a signal.

**Architecture design.** A deeper understanding of how neural networks represent implicitly encoded signals may also provide greater insights into the design of non-MLP network architectures, enabling HOSC to fully leverage their capabilities. The research presented in Ramasinghe & Lucey (2021) offers intresting ideas relevant to this context. Furthermore, HOSC naturally fits in the Factor Fields framework (Chen et al., 2023b). More specifically, we can let any factor s in a Factor Field be modeled with a HOSC MLP. This means that the framework could potentially be used to develop novel representation methods using HOSC, possibly combined with other architectures.

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

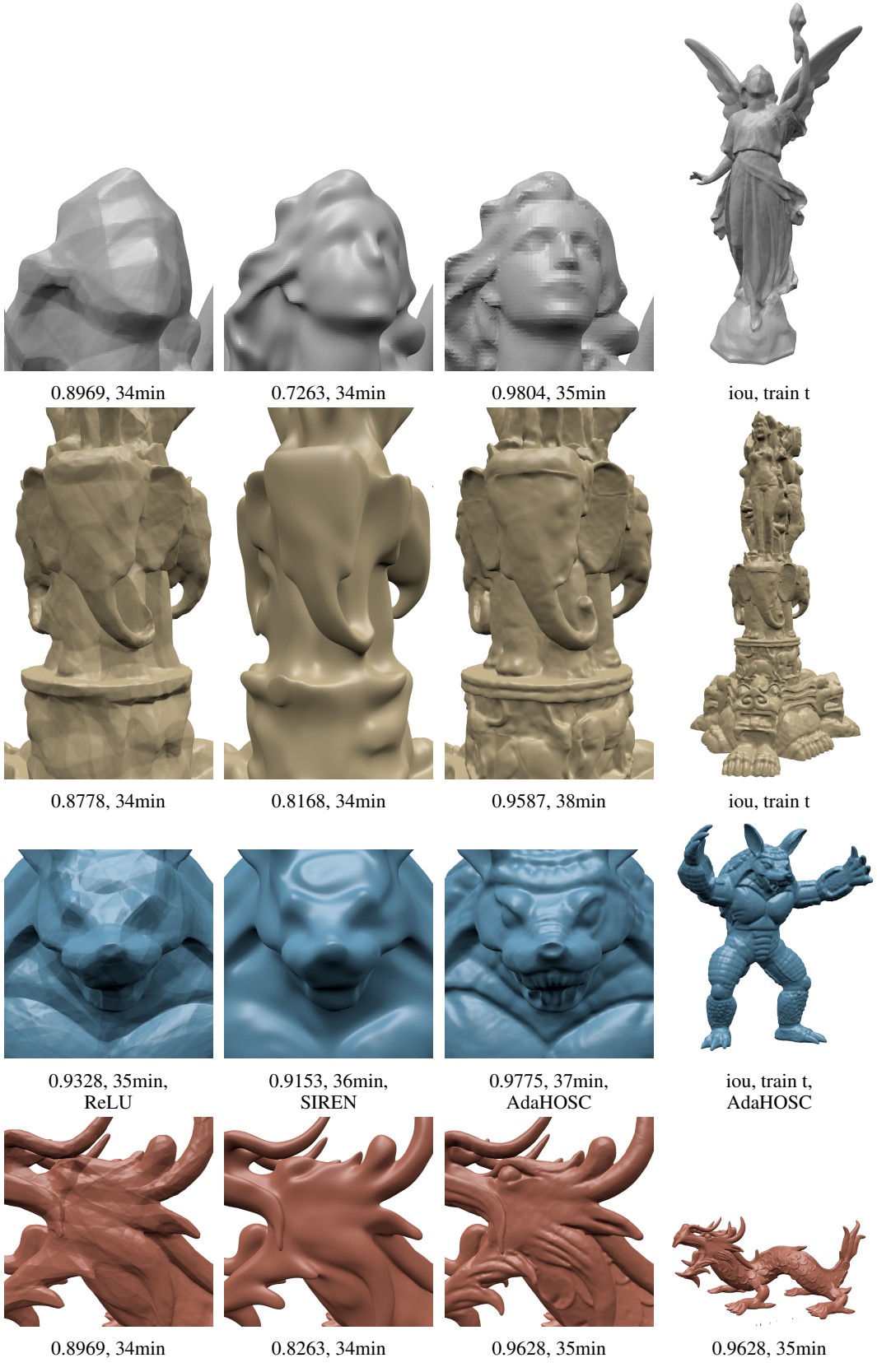

0.8969, 34min   0.7263, 34min   0.9804, 35min   iou, train t

0.8778, 34min   0.8168, 34min   0.9587, 38min   iou, train t

0.9328, 35min,   0.9153, 36min,   0.9775, 37min,   iou, train t,
ReLU             SIREN             AdaHOSC          AdaHOSC

0.8969, 34min   0.8263, 34min   0.9628, 35min   0.9628, 35min

Figure 8: Comparison of HOSC to other methods in a 3D SDF reconstruction. Training and inference time is comperable, however, the reconstruction quality of HOSC is superior, and lies in the range of methods utilizing positional encoding, like Dictionary Fields Chen et al. (2023b). For the evaluation of the IoU we used the dataset from https://github.com/autonomousvision/factor-fields. Note: lucy was evaluated only on a 512 mesh.

## A   FURTHER EXPERIMENTAL RESULTS

In this work, we introduced HOSC — a novel activation function designed specifically to preserve sharp features in implicit neural representations. In this section, we are going to present further experimental results, evaluating the performance of HOSC MLPs (MLPs employing HOSC as their activation function) in fitting synthetic signals, including stochastic superpositions of sine, square, and sawtooth waves, and Bezier curves, in comparison to ReLU and the sine activation used by the SIREN architecture, and analyze the influence of the HOSC sharpness parameter $\beta$.

### A.1   FITTING 1D SIGNALS ($\mathbf{s} \colon \mathbf{R} \to \mathbf{R}$)

We start by showing that very simple HOSC MLPs are capable of approximating waves with acute features, like the sawtooth and square waves, with much higher accuracy than ReLU MLPs or SIREN. To this end, for each of these activations we train single hidden layer MLPs of width $10$ and compare the results, depicted in Figure 9. The training set consists of $1000$ points $x$ evenly spaced in the $[-2\pi, 2\pi]$ interval along with the evaluations of the signal $s(x)$ at these points. It is observed that unlike ReLU and SIREN, a HOSC-MLP is able to accurately represent sharp jumps in the signal.

In the remaining parts of this section, we more quantitatively assess performance of HOSC in representing one dimensional signals. Thanks to their simplicity, these signals provide a versatile and interpretable benchmark. Different signals allow to stress different capabilities of HOSC, offering a comprehensive evaluation and insight into its behavior. Below, we present the results of three such experiments, highlighting the HOSC's representational capacity in comparison to the SIREN activation $\sin(x)$. We find that oftentimes HOSC outperforms SIREN, and allows the network to accurately represent more complex signals.

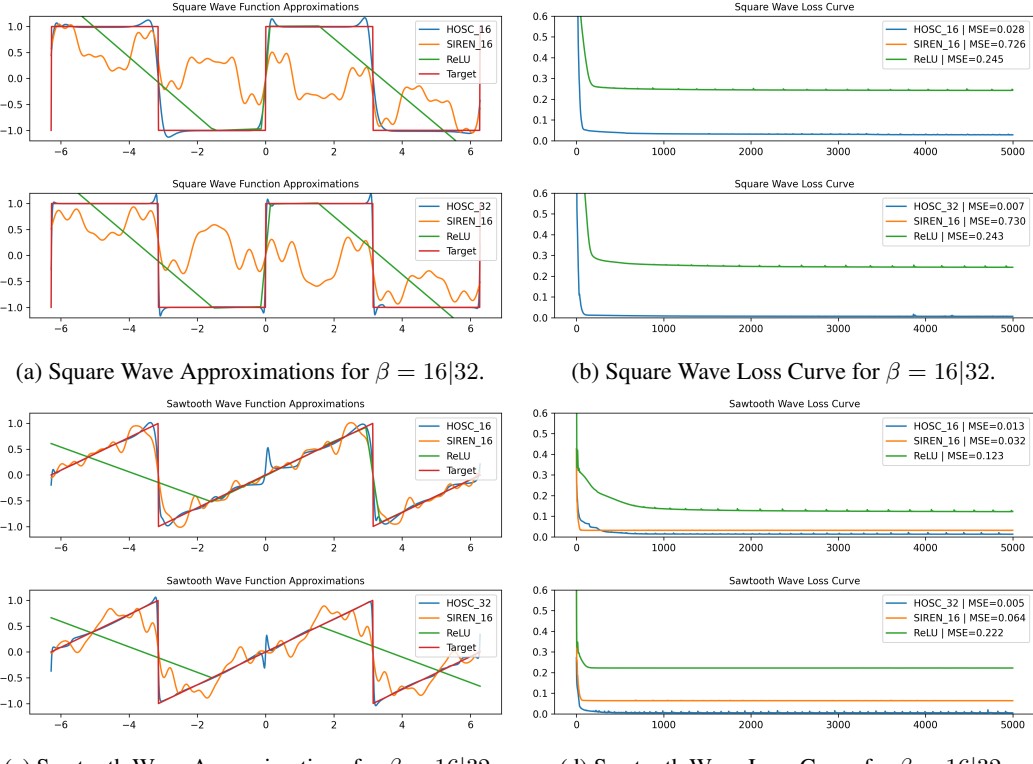

(a) Square Wave Approximations for $\beta = 16|32$.      (b) Square Wave Loss Curve for $\beta = 16|32$.

(c) Sawtooth Wave Approximations for $\beta = 16|32$.      (d) Sawtooth Wave Loss Curve for $\beta = 16|32$.

Figure 9: Comparison of ReLU, SIREN, and HOSC MLPs in fitting the sawtooth and square waves. For HOSC, we use the sharpness factors of $\beta = 16$ and $\beta = 32$. Moreover, for the SIREN activation, we used a constant frequency parameter of $\omega = 16$. On the left, the approximations approximations are displayed, while on the right we see the MSE curves.

**Experimental setup.** In each experiment, we use a two-layer multi-layer perceptron (MLP) with the hidden layer width equal to 128. The MLPs are trained on synthetic signals in the $[-1, 1]$ domain, and the results are measured using the MSE (mean square error) performance metric. Note that in these experiments we also consider different activation frequency values, i.e. we consider $\text{HOSC}(\omega x; \beta)$ and $\sin(\omega x)$ for different values of $\omega \in \mathbf{R}$.

### A.1.1 Experiment 1

The first experiment focuses on the frequency parameter $\beta$. In this experiment, we are training the MLPs on varying signal and activation frequencies, keeping the HOSC sharpness parameter constant.

**Methodology.** The training dataset consists of samples from the sawtooth and square wave signals. We allow signal and activation frequencies to range from 1 to 31 (incremented by 5). The sharpness parameter $\beta$ of HOSC is kept constant at $\beta = 1$. For each pair of parameters, we are training an MLP using HOSC and SIREN activations. Consequently, the outcome are two pairs of $7 \times 7$ matrices, one for HOSC, and one for SIREN, whose values are the mean-squared error (MSE) of the trained MLP, represented as heatmaps. The results are depicted in Figure 10.

**Conclusions.** We find that, unlike SIREN, an MLP using the HOSC activation with properely adjusted frequency, is able to learn more complex, higher frequency signals. In case of the sawtooth wave, while a SIREN-MLP is unable to achieve low MSE for signals with frequencies higher than 6, HOSC is able to accurately represent signals with frequencies as high as 26. The results for the square wave are similar, and also highlight the HOSC's capacaity to approximate signals of higher frequency than SIREN.

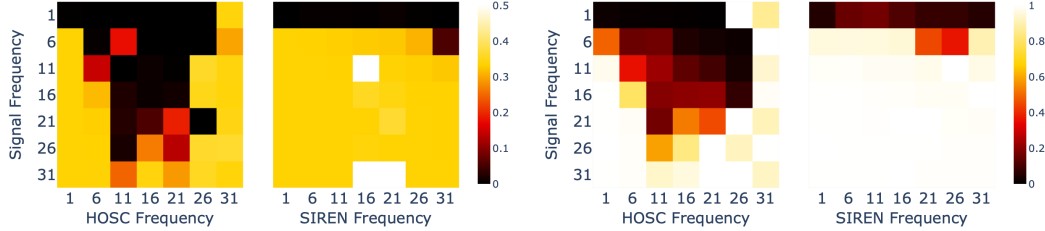

(a) MSE heatmaps for the sawtooth wave signal.  (b) MSE heatmaps for the square wave signal.

Figure 10: MSE heatmaps for HOSC and SIREN with varying signal sawtooth frequency (horizontal axis) and activation function frequency (vertical axis) parameters. Tiles with deep red and maroon hues correspond to lower MSE values, whereas tiles in shades of pale yellow and orange signify higher MSE.

### A.1.2 Experiment 2

In the second experiment, we analyze the effect of the sharpness parameter $\beta$ of HOSC on its performance. For different activation frequency values, we measure the MSE while the signal frequency and the sharpness $\beta$ vary.

**Methodology.** Again, we assess the performance on sawtooth and square wave signals. We let signal frequencies range from 1 to 36 (incremented by 5), and the sharpness from 1 to 26 (also incremented by 5). The frequency is kept constant, which yields $8 \times 6$ outcome MSE heatmap. We repeat the experiment for different values of the HOSC frequency parameter: $[1, 2.5, 5, 7.5, 10, 20]$, which results in six heatmaps, depicted in Figure 11.

**Conclusions.** We conclude that fine-tuning the HOSC sharpness parameter $\beta$ results in better aproximation of the signal, especially in the case of the sawtooth wave. However, this applies only when the frequency $\omega$ is kept low, with the best results achieved around $\omega = 5$. For $\omega = 20$ increasing the sharpness does not have a positive effect, and in fact hinders learning.

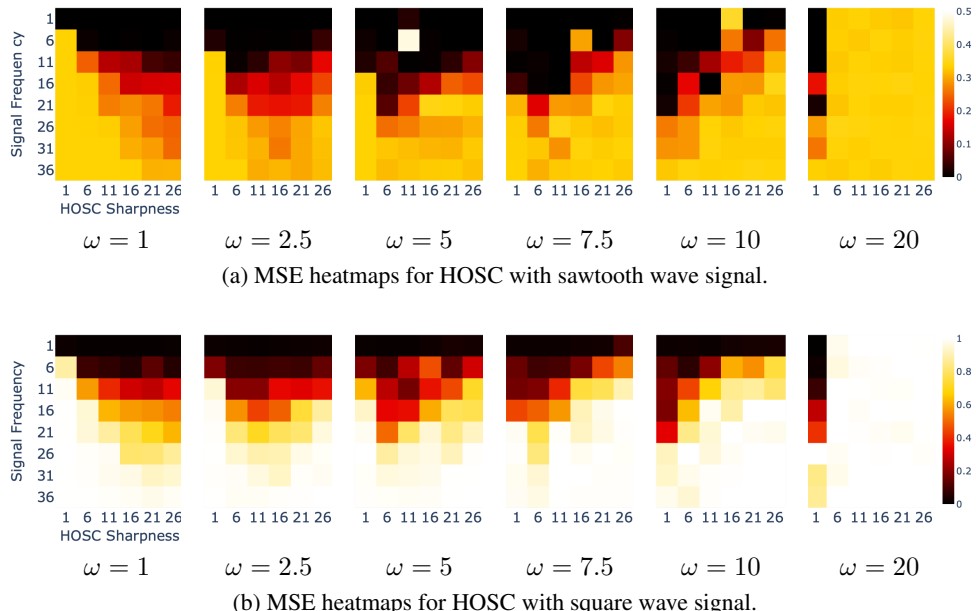

(a) MSE heatmaps for HOSC with sawtooth wave signal.

(b) MSE heatmaps for HOSC with square wave signal.

Figure 11: Performance analysis of HOSC focusing on the sharpness parameter $\beta$. Six MSE heatmaps are displayed for each of the sawtooth and square wave signals, corresponding to different values of the frequency parameter $\omega$. Each heatmap varies the signal frequency from 1 to 31 and the HOSC sharpness factor $\beta$ from 1 to 26. Lower MSE values are indicated by deep blue and purple hues, while higher MSE values are shown in green and yellow. Results suggest that fine-tuning $\beta$ improves signal approximation, particularly for the sawtooth wave, when the frequency $\omega$ is low.

### A.1.3 EXPERIMENT 3

In the third experiment, we analyse the joint effect of the frequency and sharpness parameters of HOSC. We assess the performance both in isolation and against the SIREN activation $\sin(x)$. The experiment uses a random signal constructed as a sum of randomly weighted sines, which allows to measure the robustness in representing diverse signal features.

Table 1: MSE achieved by MLPs using HOSC with parameters $\beta$ and $\omega$ on random signal benchmark in comparison with the SIREN acitvation.

| | HOSC sharpness $\beta$ | | | | | | |
|---|---|---|---|---|---|---|---|
| $\omega$ | 1.0 | 6.0 | 11.0 | 16.0 | 21.0 | 26.0 | SIREN |
| 1.0 | 0.023706 | **0.004319** | **0.004811** | 0.020108 | 0.056000 | 0.122588 | 0.015133 |
| 6.0 | **0.000524** | **0.008368** | **0.028785** | 0.101580 | 0.127431 | 0.126491 | 0.094149 |
| 11.0 | **0.000910** | **0.008134** | **0.043188** | 0.124307 | 0.121590 | 0.129529 | 0.078931 |
| 16.0 | **0.002684** | **0.009155** | 0.098988 | 0.127735 | 0.124204 | 0.123641 | 0.050414 |
| 21.0 | **0.003916** | **0.019826** | 0.117844 | 0.124351 | 0.125705 | 0.126329 | 0.055959 |
| 26.0 | **0.012180** | **0.027917** | 0.122362 | 0.131056 | 0.122926 | 0.124286 | 0.051204 |

**Methodology.** We start by defining the algorithm generating random signals $\xi\colon [-1, 1] \to \mathbf{R}$. They are constructed as a probabilistically weighted sum of sines with random frequencies $\omega_i$ and a random offset $\pi_i$. Formally, we let

$$\xi(x) = \sum_{i=1}^{n} \lambda_i s(\omega_i x + \pi_i),$$

where $s(x) = \sin(x)$, $\|\lambda\|_1 = 1$, $\lambda_i > 0$, and $\pi_i \in [0, 2\pi]$ for all indices $i$. Moreover, we choose the number of components $n = 100$ and constrain the frequencies to $\omega_i \in [0, 100]$ to lie in the $[0, 100]$

interval. The final random signal $\xi^*\colon [-1, 1] \to \mathbf{R}$ is obtained by normalizing $\xi$ in the $L_\infty$-norm: $\xi^* = \xi/\|\xi\|_\infty$. Some examples of random signals constructed using this method (as well as their approximations) are depicted in Figure 13.

We let the HOSC sharpness and frequency parameters range from 1 to 26 (incremented by 5), and sample 100 random signals. We train a separate MLP on each one of them, and average the loss to create a $6 \times 6$ output MSE heatmap, as depcited in Figure 12. Moreover, we compare the results to the MSE achieved by SIREN on the identical task. In the table 1, we include the exact MSE values achieved by HOSC in this experiment, and bold the ones where HOSC outperformed SIREN.

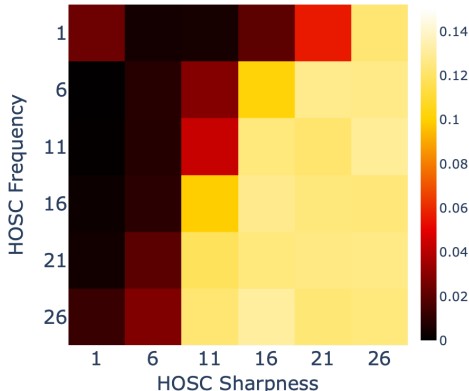

Figure 12: MSE heatmap for HOSC applied on random weighted-sum-of-sines signals with varying frequency (vertical axis) and sharpness (horizontal axis) parameters. Tiles with darker hues correspond to lower MSE values.

**Conclusions.** The HOSC activation allows MLPs to closely approximate random signals constructed as weighted sums of sines, especially with fine-tuned parameters $\beta$ and $\omega$. MLPs with SIREN activation are unable to achieve similarly low MSE.

## A.2   Fitting Bezier curves ($\mathbf{s}\colon \mathbf{R} \to \mathbf{R}^n$)

It is worthwhile to examine the behavior of fitting INRs to a series of sampled signals of ascending total curvature. To this end, we fit four-layers 256 width MLPs running ReLU, SIREN, and HOSC activations to a randomly sampled Bezier curve

$$\mathrm{B}_{(m,d)}(x) = \sum_{k=1}^{m} \binom{k}{m}(1-x)^{m-k}x^k \cdot \mathbf{p}_k\,,$$

where $\mathbf{p}_k \sim \mathrm{N}(\mathbf{0}, \mathbf{I}^d)$ is randomly sampled from the standard normal $d$-dimensional distribution. In the experiment, we let $m \geq 4$, as then we are guaranteed that $\mathrm{B}_{(m,d)}$ is twice-differentiable:

$$\mathrm{B}''_{(m,d)}(x) = m(m-1)\sum_{k=0}^{m-2}\binom{k}{m-2}(1-x)^{m-k-2}x^k(\mathbf{p}_{i+2} - 2\mathbf{p}_{i+1} + \mathbf{p}_i)\,.$$

This implies that the total curvature of $\mathrm{B}_{(m,d)}$ increases as the number of control points $m$ and the dimension $d$ increase. For HOSC, we train two MLPs: the first one, denoted **hosc**, uses a scheduled sharpness schedule $\beta_i \in [2, 4, 8, 16]$; the second one, **hosc_all_8** uses a constant sharpness $\beta = 8$. The results are displayed in Figure 14. We observe that while ReLU-MLP's and SIREN's PSNR values start to decrease as the number of control points $m$ increases, wheras both HOSC-MLP models seem to consistently achieve high PSNR.

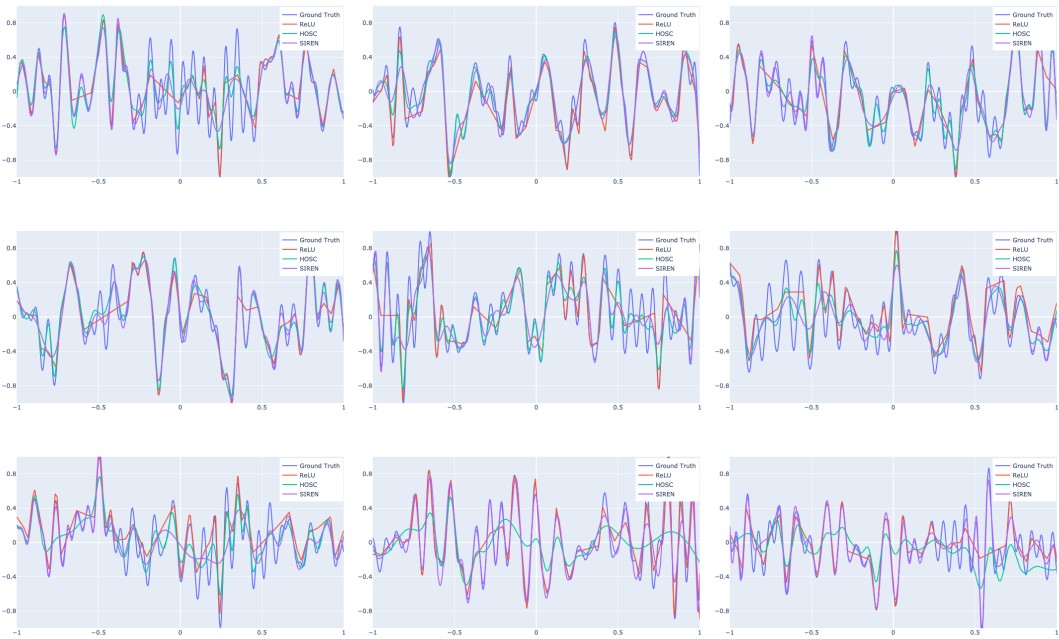

Figure 13: Examples of stochastic signals and their approximations using an MLP running ReLU, HOSC, and SIREN activations. Oftentimes an MLP with HOSC activation is able to accurately capture the signal's features, while ReLU and SIREN are unable to do so. Last two plots depict a situation where HOSC fails to learn a representation of the signal.

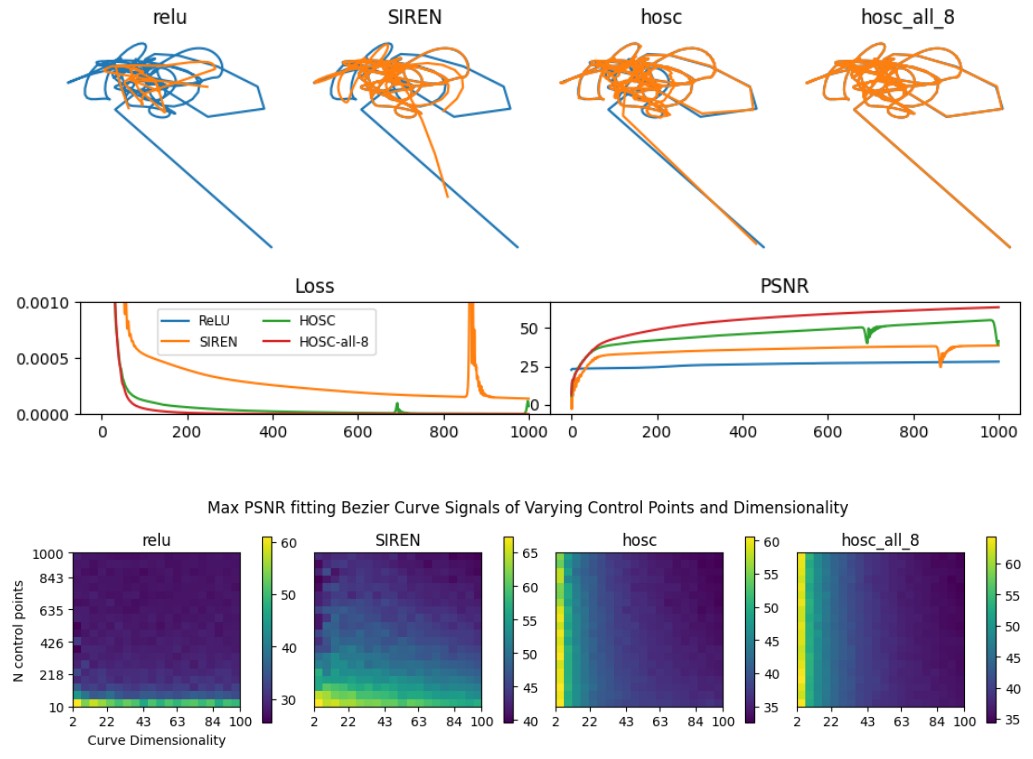

Figure 14: Comparison of ReLU, SIREN, and HOSC-MLPs fitting a 2D Bezier curve with $m = 1000$ randomly placed control points for 1000 epochs. Top: Bezier curves approximations constructed with different models. Bottom: PSNR heatmaps for different values of dimensionality $d$ and numbers of control points $m$.

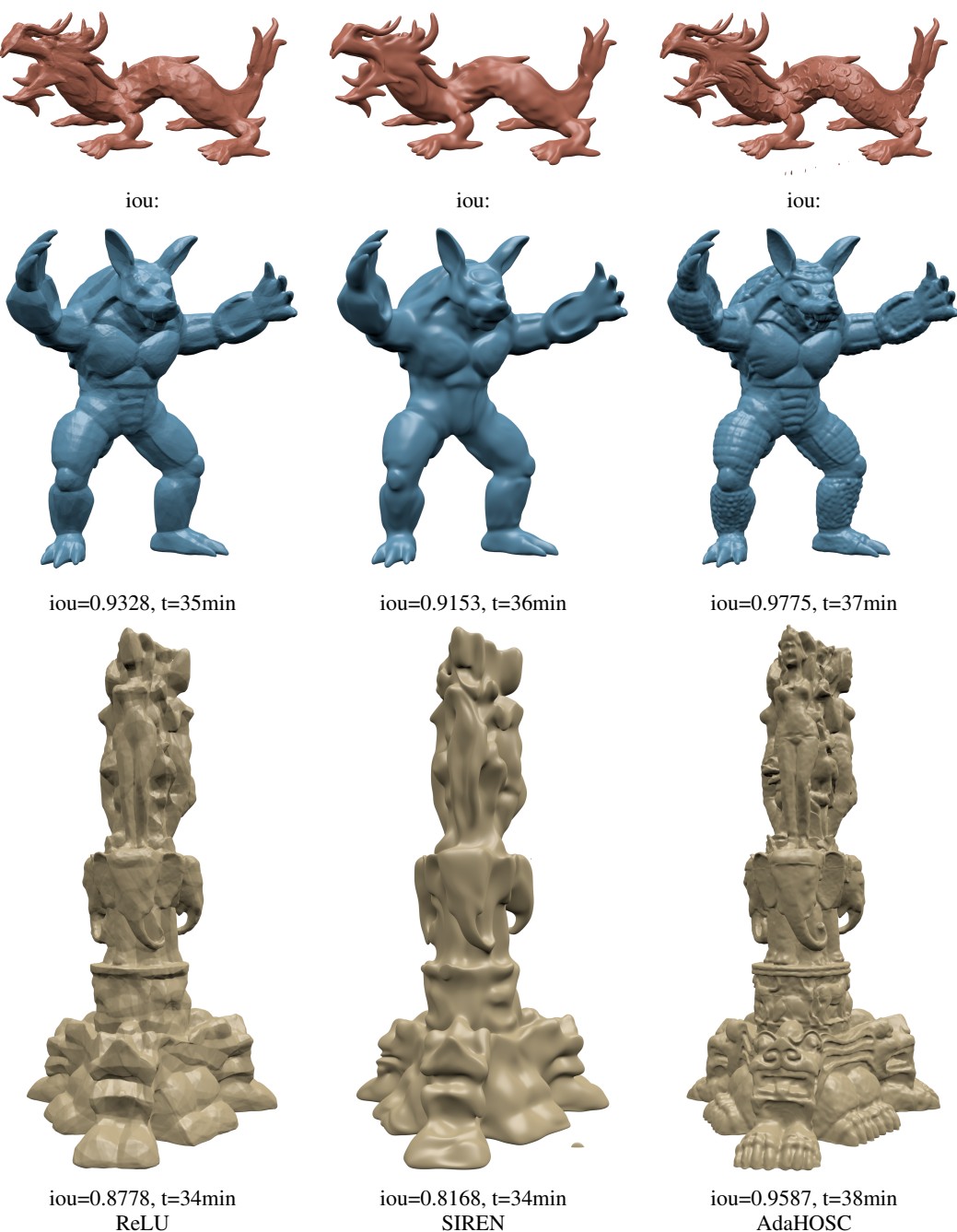

iou:                          iou:                          iou:

iou=0.9328, t=35min          iou=0.9153, t=36min          iou=0.9775, t=37min

iou=0.8778, t=34min          iou=0.8168, t=34min          iou=0.9587, t=38min
ReLU                          SIREN                        AdaHOSC

Figure 15: Further results of the 3D SDF experiment.

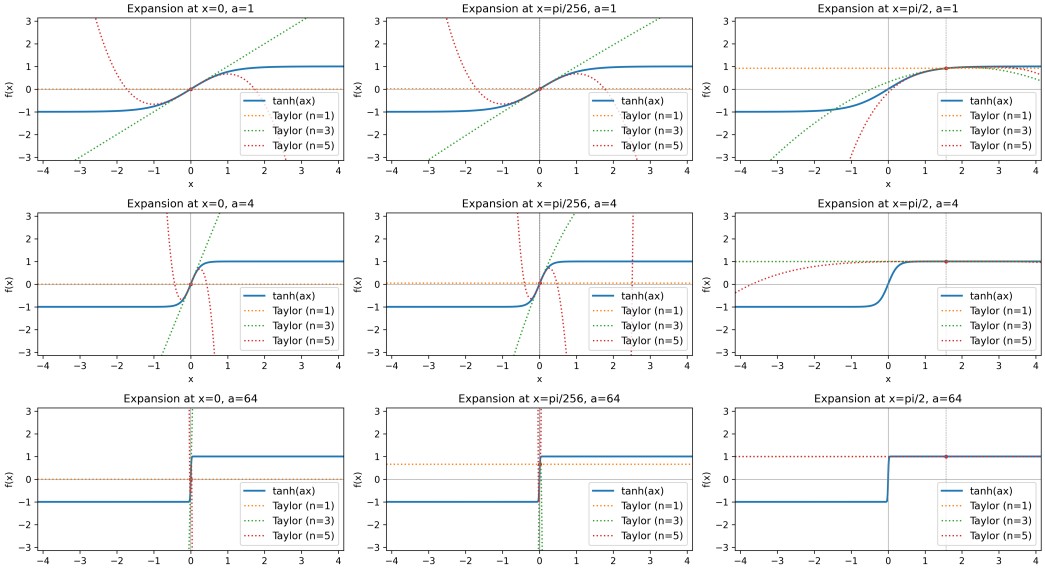

Figure 16: Taylor series approximation of $\tanh(ax)$. The parameter $a$ controls the steepnes of the transition.

## B   THEORETICAL ANALYSIS OF HOSC

In this section, we discuss mathematical motivaitons behind the HOSC function, defined as

$$\text{HOSC}(x; \beta) = \tanh(\beta \sin x).$$

### B.1   TAYLOR APPROXIMATION

Recall the hyperbolic tangent function $\tanh x = (e^x - e^{-x})/(e^x + e^{-x})$. In Figure 16 the Taylor series approximation of $\tanh(ax)$ at different points $x$ for $a \in \{1, 4, 64\}$. At $x_0 = 0$, we have

$$\tanh(x) = x - \frac{x^3}{3} + \frac{2x^5}{15} - \frac{17x^7}{315} + \frac{62x^9}{2835} + O\left(x^{11}\right)$$

This expansion reveals the underlying structure of the $\tanh(ax)$ function, highlighting its behavior for varying values of $x$. For small values of $x$, the function behaves nearly linearly, and as $x$ increases, the higher-order terms contribute to a steeper transition, eventually approaching the horizontal asymptotes at $\pm 1$.

In the context of the HOSC function, the $\tanh(x)$ component serves to modulate the sine wave, effectively controlling the transition between values. Steeper transitions allow HOSC to preserve edges and corners.

### B.2   SPECTRAL ANALYSIS

The HOSC function can be represented as a Fourier series

$$\sum_{n=0}^{\infty} b_n \sin(nx)$$

with

$$\frac{\pi}{2} \int_{-\pi}^{\pi} \text{HOSC}(x; \beta) \sin(nx) \, \mathrm{d}x.$$

Since the function is point-symmetric, only $b_n$ coefficients are needed, and all $a_n$ including $a_0$ cancel out. Due to the complexity of the $\tanh x$ function, deriving the Fourier coefficients analytically is challenging, and we resort to numerical integration, leaving the analytical solution for future work.

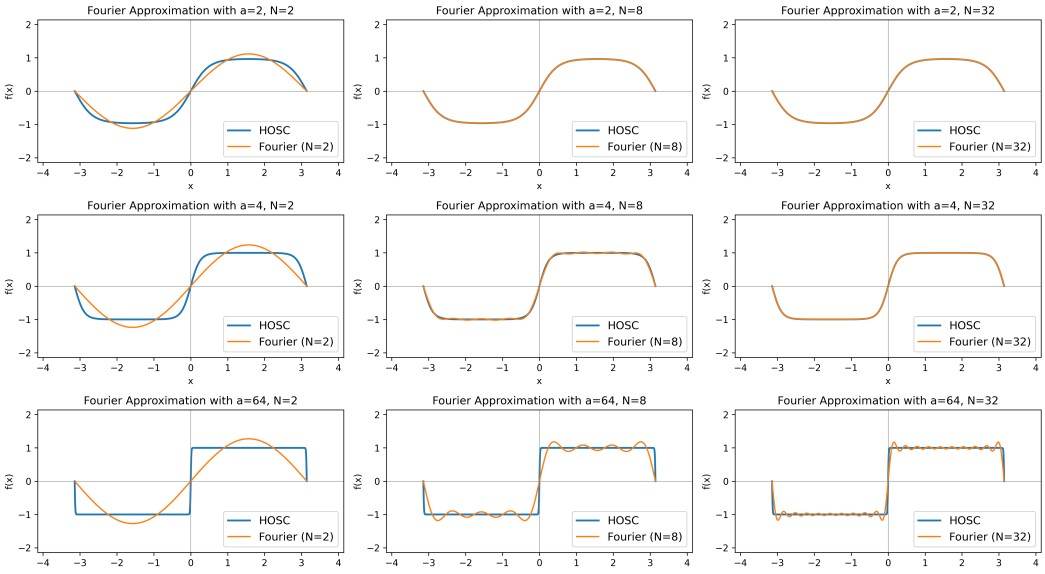

Figure 17: Fourier series approximation of the HOSC function for different values of $\beta$. The plots illustrate the relationship between the sharpness factor and the number of Fourier terms required for approximation. As $\beta$ increases, more terms are needed to capture the function's behavior, reflecting the transition from sine-like behavior to more complex oscillations.

The Fourier series representation provides insight into the frequency characteristics of an activation function. In Figure 17, we present a comparison of the frequency response of HOSC with the SIREN activation. HOSC's Fourier coefficients exhibit a different distribution compared to SIREN, which is particularly evident in the high-frequency components.

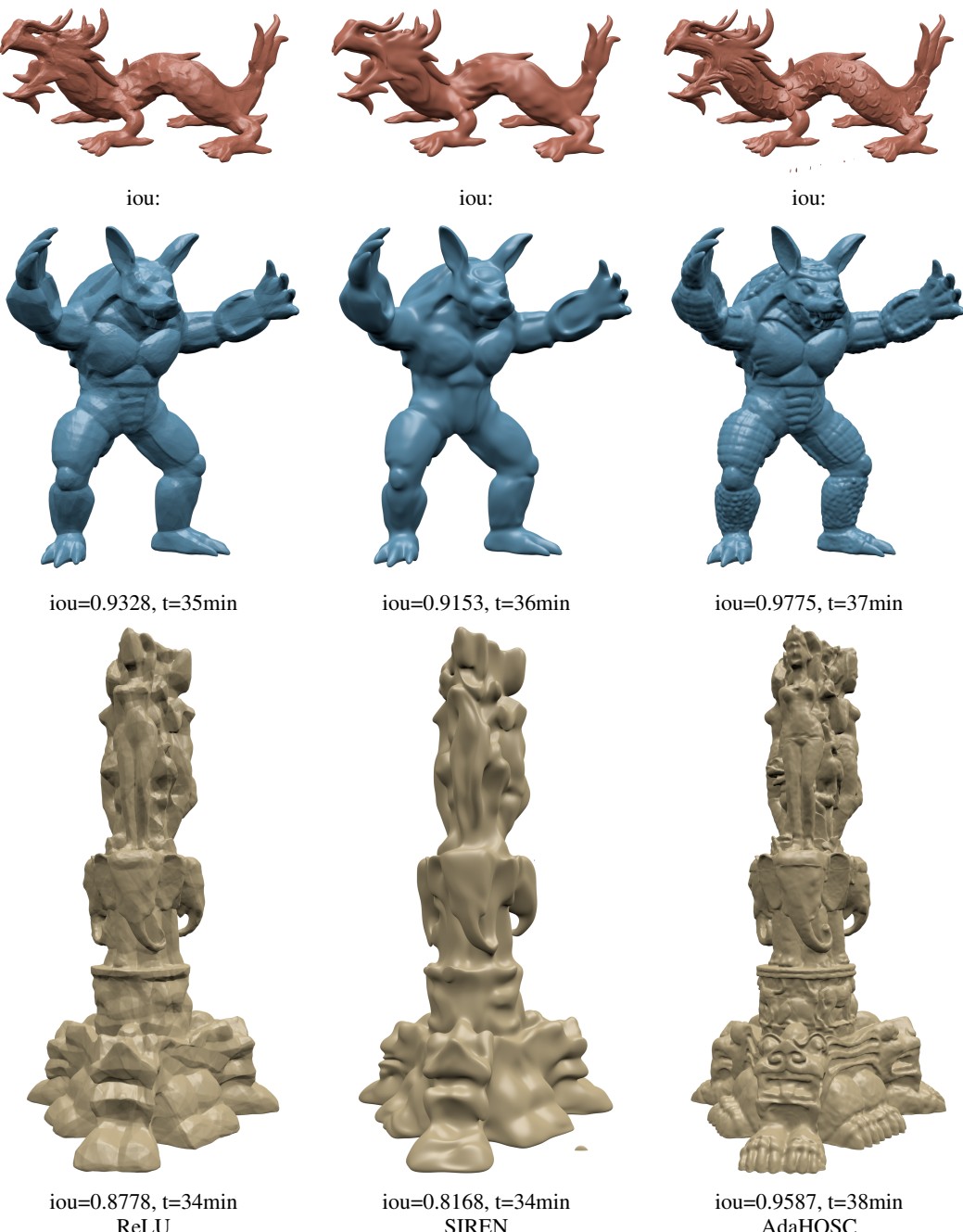

iou:                                iou:                                iou:

iou=0.9328, t=35min      iou=0.9153, t=36min      iou=0.9775, t=37min

iou=0.8778, t=34min      iou=0.8168, t=34min      iou=0.9587, t=38min
ReLU                              SIREN                           AdaHOSC

Figure 18: Further results of the 3D SDF experiment.

