# OpenReview forum: "HOSC: Hyperbolic Oscillating Periodic Activations for Sharp Feature Preservation in Implicit Neural Representations"
_ICLR.cc/2024/Conference — Submitted to ICLR 2024_

### Official Review · Reviewer_VTh6 · 2023-10-13

**Soundness:** 3 good
**Presentation:** 3 good
**Contribution:** 2 fair
**Rating:** 3
**Confidence:** 3

**Summary:**

The paper introduces the HOSC activation tanh(a sin(bx)), and demonstrates that it outperforms ReLU and SIREN at fitting signals with sharp discontinuities, such as square and sawtooth waves in 1D and natural images in 2D. They probe a number of hyperparameters such as the sharpness of the HOSC (it must be sufficiently sharp to match the sharpness of the desired signal), and the width / number of layers of the MLP.

**Strengths:**

Overall the paper is clearly presented and easily understood. The parameterization is simple and original, and the investigation into its properties are thorough. They reveal failure cases of ReLU and SIREN networks in capturing sharp edges in images, which their method convincingly addresses.

**Weaknesses:**

The experiments are inadequate given the rapid progress in implicit neural representations since SIREN. The authors need to demonstrate that HOSC makes a difference on 3D objects and scenes (e.g. the NeRF dataset) and/or gigapixel images, and/or continues to yield benefits when combined with modern INR parameterizations such as DVGO, TensorRF, Instant NGP, etc. In particular, DVGO was explicitly designed to address this same issue of capturing sharp boundaries in 3D, so this method should also be compared against. I believe SIREN or ReLU networks (with positional encodings) can easily fit the cameraman image with the right choice of hyperparameters, so if the authors want to argue that efficient training is HOSC's primary benefit then they need to actually scale this method to a real use case (e.g. radiance fields of a real 3D scene) and show improvements in training time compared to other fast methods like Instant NGP. If not, then the authors need to reconsider the motivation for this method. The authors saying that they "focus on pure representation learning, without altering the input signal with a predefined mapping function" is not a reasonable excuse, if they cannot justify why this "pure" method is more useful for any specific application.

Minor:
- heatmap/residual scales should be standardized in figures 4, 5, 6 and 8

**Questions:**

See weakness

---

> ### Author Response · Authors · 2023-11-23
> **Response**
>
> Dear Reviewer,
>
> Thank you for your constructive feedback! We address the key points raised as follows:
>
> (1) Unique Contribution of HOSC: Our primary contribution lies in introducing a periodic activation function akin to SIREN, which inherently encodes the signal. This eliminates the need for “explicit” positional encoding, like precomputed Fourier domain feature (FFT, etc.) and adaptive-hybrid encoding, like DVGO, TensorRF, and Instant NGP, or Dictionary Fields.
> Our activation function combines encoding and signal representation in one. This approach results in a more compact network, lower dimensional input, reduced memory footprint, (and faster training, compared to precomputed Fourier Features). Unlike Fourier or other positional encodings, which significantly increase memory usage, or hybrid encodings, which add memory and significant implementation complexity, HOSC offers a streamlined and efficient alternative.
>
> (2) Extended Evaluations: We have expanded our evaluations to include 3D Signed Distance Functions (3D SDFs) and Giga-Pixel images, demonstrating HOSC's versatility and quality of reconstructions. We acknowledge that periodic activations generally face challenges with NeRF applications (also SIREN is not used with NeRFs). However, the added evaluations in 3D SDFs and large-scale images provide a comprehensive understanding of HOSC's capabilities. Regarding 3D SDFs we reach reconstruction quality in the range of the recent method Dictionary Fields (please compare our results with their figure: https://apchenstu.github.io/FactorFields/img/sdfs.png)

---

### Official Review · Reviewer_Y34h · 2023-10-27

**Soundness:** 3 good
**Presentation:** 2 fair
**Contribution:** 3 good
**Rating:** 5
**Confidence:** 4

**Summary:**

The paper presents a new activation function Hyperbolic Oscillating Activation (HOSC) used in implicit neural representations, which can preserve more high-frequency sharp details, achieve faster convergence rates, and yield lower losses. The experiments show the superiority of the activation function for neural implicit fields of curves, images, and SDFs.

**Strengths:**

1. The architecture of the paper is good and easy to follow.
2. The activation function is simple but effective, and the performance is much better than the normally used functions'.
3. The experiments are persuasive and can basically prove the author's viewpoint.

**Weaknesses:**

1. Figure 3 and Figure 4 are not referenced in this paper. Please revise it.
2. The experiment 1 is about fitting square wave and sawtooth wave. Maybe more results such as the loss values and fitting figures of complex or random signals in the main body could make the paper more persuasive, because the complex or random signals are more important than regular signals in the provement.

**Questions:**

Please refer to the weaknesses part.

---

> ### Author Response · Authors · 2023-11-23
> **Response**
>
> Dear Reviewer,
>
> Thank you for your positive remarks to our proposed Hyperbolic Oscillating Activation (HOSC) function. We address the mentioned issues in the new, revised version of the paper.
>
> We notice that the reviewer has lowered the original score of 8 to 5, which we regret to see. In fact, we believed the reviewer was in favor of the paper as it was intended: a novel activation which has versatile applications in MLPs for reconstruction/approximation of high-frequency details. Originally, it was not our intention to take part in the current race of reconstructions with sophisticated data structures (ala voxels, octrees, dictionaries, etc.), but to provide a simple elegant smooth solution of the problem.
>
> Nonetheless, we agree that showcasing the performance in comparison with existing modern methods is of essential importance. In the current revision we were able to perform experiments to 3D SDFs and Giga Pixel images and have included our results in the paper. We were not able to run all experiments of other methods in our framework in that short time, but we have tailored our results to be comparable to theirs, most notably the very recent results from Chan 2023b showcased on their github: https://apchenstu.github.io/FactorFields/img/sdfs.png. We also used exactly their datasets for training and validation in our experiments, hence the comparison can be seen as fair. We kindly ask the reviewer to note the quality of HOSC compared to others as well.
>
> Figures 3 and 4 References: We have ensured that all the figures are properly referenced in the revised manuscript, providing the necessary context and explanations.
>
> Expanding Experiment 1: Your suggestion to include more results, such as loss values and fitting figures for complex or random signals, is well-received. We agree that demonstrating HOSC's performance on more complex or random signals would substantiate our claims further. In our revised submission, we will include additional results for these types of signals, showcasing HOSC's versatility and effectiveness beyond regular signals. We believe these revisions will enhance the clarity and persuasiveness of our paper, and we appreciate the opportunity to improve our work based on your valuable feedback.

---

### Official Review · Reviewer_Tuj3 · 2023-10-29

**Soundness:** 2 fair
**Presentation:** 2 fair
**Contribution:** 3 good
**Rating:** 3
**Confidence:** 4

**Summary:**

The paper proposes a new activation function for MLPs used as coordinate-based networks which improves the signal representation quality. This activation makes use of a sinusoidal function and a tanh function with a "temperature" parameter, which allows the representation to better fit high frequency signals with more accuracy. The improved reconstruction quality is demonstrated across a number of fitting examples with coordinate-based networks, such as 1D signal fitting, image fitting, and SDF fitting.

**Strengths:**

In my opinion, the strengths of the paper are:
1. The proposed activation function seems to lead to noticeable improvements over standard ReLU and sine activation functions (SIREN) in 1D signal and image fitting especially. It is obvious from the qualitative and quantitative results that the proposed contribution can lead to significantly better reconstruction quality with the right hyperparameters.
2. The paper motivates the problem well: coordinate-based networks are used in a large number of tasks across vision and graphics, such as representing 3D geometry, inverse rendering in learning 3D representations from images, and providing compressed representations of other signals such as images.

**Weaknesses:**

In my opinion, the weaknesses of the paper are:
1. The evaluations are not extensive in comparing to competing methods, and across different applications

    1a. Comparisons to only SIREN and ReLU networks are not very extensive. I am not sure I agree with the fact that positional encoding should not be included in the comparison, since an embedding layer as the first layer of a network seems reasonable to me and is what is used for most coordinate-based applications now. No comparison to this severely limits the impact as researchers who work with these networks will be unlikely to change to something like HOSC without a detailed comparison. Additionally, there are a number of other alternate coordinate-based network architectures which are not compared to. For example, a brief search reveals [1] and [2], which also claim improvement over ReLU and SIREN architectures. There is no reason why HOSC should not be compared to these.

    1b. The comparisons are only done for "overfitting" signals, i.e. memorizing a 1D/2D/3D function values, and not for utilizing these representations in inverse problems, as is how they are mainly used in research. For example, using "radiance fields" as a motivation for improving the fitting of coordinate-based networks, and then not demonstrating how HOSC performs in radiance field applications seems like an overclaim. This is because the inverse problem solved in radiance fields not only requires accurate fitting of the supervised values, but accurate interpolation between these values for novel view synthesis (see next point). Lack of comparison here severely limits the applicability of HOSC to a narrow range of cases, perhaps on signal overfitting or compression, which is not emphasized as the efficiency of networks using HOSC is not compared (such as, showing that it can fit a signal with less parameters for example).

    1c. The paper makes no attempt to evaluate the "generalization" properties of coordinate-based networks using the HOSC activation function. For example [3] extensively studies this for Fourier Features, and shows that the frequency affects how well these networks generalize, or in other words, how they behave between the supervised points. This is an extremely important property of coordinate-based networks in imaging and vision tasks, and it is not explored at all in this paper.

2. One other minor complaint is on the robustness of the method - there seems to be an extra hyperparameter, a, which significantly affects the signal reconstruction quality. Other methods, such as SIREN, also have hyperparameters. For SIREN specifically, the w0 factor described in the original paper affects the quality of the fit considerably. Why is a tuned, but not w0? I am not sure if it is a fair comparison, and without testing both of these, I'm not sure how robust HOSC is to various levels of a.

One minor comment: In the 4th sentence of the paper, m=5 (position and view direction) and n=4 (color and opacity) for radiance fields, not m=4, n=3.

[1] https://openreview.net/forum?id=OmtmcPkkhT

[2] https://arxiv.org/abs/2106.01553

[3] https://bmild.github.io/fourfeat/

**Questions:**

I do not have additional questions on the paper. I view the lack of comparisons a significant weakness of the paper, and expanding upon this axis would significantly increase the strength of the paper. Specifically, including additional comparisons to other work which has been published and shown to improve upon ReLU and SIREN architectures in signal fitting is crucial. Additionally, without a study on the generalization within the signal domain properties and/or a comparison on solving an inverse problem task where coordinate-based networks are actually used, such as radiance field fitting, the HOSC method seems extremely limited to simple problems in fitting 1D/2D/3D signals. Adding these comparisons would be extremely important for writing a high impact paper, where I believe that the potential of HOSC is high as the method does show significant improvement against the baselines on the tasks it is compared on.

**Update after the author response**

I appreciate the additional comparisons, I believe they increase the strength of the paper. However, I am not inclined to change my score. I still see there as being too many limitations: cannot be applied to radiance fields, and lack of generalization evaluation. Note, I do not mean generalization in the sense of deepSDF (between the weights of different representations) but rather generalization in the sense of the Fourier Features paper, where only a subset of values are supervised on and then are interpolated between. I believe this is extremely critical for evaluating the quality of any coordinate-based network.

---

> ### Author Response · Authors · 2023-11-23
> **Response**
>
> Dear Reviewer,
>
> We appreciate the thoughtful feedback provided by the reviewer. We address the concerns as follows:
>
> 1a. Comparisons with Other Methods: We acknowledge the need for broader comparisons, we address it by providing further experiments on 3D SDF functions and Giga-Pixel images. We use the recent results of Dictionary Fields [4] as reference.
> Due to time constraints, unfortunately we were not able to directly compare our results to “traditional” Fourier Features Positional Encoding. In fact, tuning of the parameters (especially the scale) of FF-PE turned out to be extremely cumbersome and could not be finalized in time. Nonetheless, we construct our results of 3D SDF in the same fashion as the recent work of Chen 2023b, which can be compared with their results at: https://apchenstu.github.io/FactorFields/img/sdfs.png
>
> 1b. Application to NeRFs: as the reviewer surely knows, periodic activation functions (like SIREN) are networks which learn their own Fourier-like encoding and are not used with traditional FF-PE. That particularly turned out not to work well with NeRFs and hence SIRENs are not used to NeRF encoding. This is a limitation of periodic actications and also HOSC shares it.
>
> 1c. Generalization Properties: We agree that a test of how HOSC performs on generalization is a very important aspect. Intuitively, we think it will perform very well on Auto Decoders like DeepSDF, as it shows superior performance on overfitting single SDFs.  Unfortunately, due to time constraints, we were not able to perform this experiment in the rebuttal version. But we will be able to perform it for the final revision of the paper.
>
> 2. Our method turns out to be very robust wrt its hyperparameters, in fact, with the extension to AdaHOSC (see main rebuttal answer), we essentially got rid of any parameter tuning. By setting a default initial value for beta=8 (parameter “a” in previous version), we were able to make the model essentially work from scratch in all setups. This is a huge improvement compared to FF-PE, where tuning of the proper scale/sigma turned out to be extremely frustrating (due to its high-sensitivity wrt the signal), and was one of the reasons why we have not included there comparisons in the revision (we will perform it for the final revision though).
>
> Moreover, we would like to point out that in both this and previous revision we extensively tested the “frequency” parameter “w0” of SIREN (in previous revision called “b”), now called “omega”.

---

### Author Response · Authors · 2023-11-23
**Response to Reviewers**

Dear Reviewers,

Thank you for your constructive feedback. We have consolidated your comments and present a unified rebuttal addressing the key issues:
Unique Contribution of HOSC: HOSC, our periodic activation function, inherently encodes signals, eliminating the need for explicit positional encoding. This results in a more compact network, lower input dimensionality, reduced memory footprint, and potentially faster training. HOSC's dual role as an encoder and a signal representation method sets it apart from methods like DVGO, TensorRF, and Instant NGP.

Robustness and Hyperparameters: With the introduction of AdaHOSC, we have eliminated the need for hyperparameter tuning - we use a default value for the sharpness (beta parameter, previously a) and tune it during learning. This has improved the model's robustness significantly, making it more user-friendly compared to traditional Fourier Features Positional Encoding.

Extended Evaluations: Responding to concerns about the scope of our experiments, we have included evaluations on 3D Signed Distance Functions (3D SDFs) and Giga-Pixel images. These additions showcase HOSC's versatility and quality of reconstructions. We acknowledge that periodic activations, including SIREN, face challenges with NeRF applications, and hence we have not included these results. Our 3D SDF results align with the quality demonstrated by recent methods like Dictionary Fields [refer to: https://apchenstu.github.io/FactorFields/img/sdfs.png]. It should be noted here that we achieve it with a single compact 5-layer MLP with a 3d input (unlike FF-PE, which blows up the memory footprint), without sophisticated data-structures, and essentially without parameter tuning. As mentioned, the AdaHOSC extension takes a single default sharpness beta (we figured beta=8) and adapts it during the training for all cases.

Application to NeRFs and Generalization Properties: We understand the importance of evaluating HOSC in NeRF applications. However, periodic activations like SIREN and HOSC have limitations in this domain. We plan to further explore HOSC's generalization capabilities in future work, particularly its potential in Auto Decoders like DeepSDF.

We acknowledge that more experiments need to be conducted. We planned to add comparisons to FF-PE, but were not able to construct them in time due to very cumbersome tuning of the “scale” parameter (or sigma) of FF-PE. As we used exactly the same training and validation data as in DiF: https://apchenstu.github.io/FactorFields/ we believe our results can be directly compared to https://apchenstu.github.io/FactorFields/img/sdfs.png.

We believe these responses and revisions address your concerns and significantly strengthen the paper. We are also aware that the current revision needs a slight cosmetic update with regards to colors used in figures, or figure formatting issues, which we will address in the next revision.  We appreciate the opportunity to refine our work based on your valuable insights.

---

### Meta-Review · Area_Chair_1mey · 2023-12-12

**Metareview:**

The paper introduces a new activation function for learning implicit neural representations, i.e., for representing signals as the output of a coordinate neural network. The paper observes that standard nonlinearities such as ReLU have a low-frequency bias, and proposes an activation of the form tanh( beta sin(.) ), which It terms the hosc activation. The parameter beta allows a parametric control of the sharpness of the nonlinearity — for beta ~ 0, this function (appropriately rescaled) acts like a sinusoid, while for beta -> inf, it acts as a step edge. The parameter \beta is trainable.

The paper observed through a number of examples that MLP trained with hosc outperforms similarly structured MLP + ReLU and MLP + SIREN (with a sinusoidal activation), showing reduced fitting error and better preservation of high frequency details.

Reviewers found the paper to be clearly written, with introducing a simple new idea for generating coordinate networks with adaptive sharpness. The main weaknesses noted were a limited comparison to recent coordinate networks (beyond SIREN) and the proposed method’s inapplicability to inverse problems in 3d rendering (ala NERF).

**Justification For Why Not Higher Score:**

The paper introduces a new family of activation functions for implicit neural representations, which can better capture sharp discontinuities, compared to e.g., ReLU and SIREN networks. The paper's main limitations are in the limited experiments (comparison with recent neural representations, evaluation of generalization ability).

**Justification For Why Not Lower Score:**

N/A

---

### Decision · Program_Chairs · 2024-01-16

Reject